# Evolutionary adaptation after crippling cell polarization follows reproducible trajectories

**Liedewij Laan\*†, John H Koschwanez, Andrew W Murray**

FAS Center for Systems Biology, Department of Molecular and Cellular Biology, Harvard University, Cambridge, United States

**Abstract** Cells are organized by functional modules, which typically contain components whose removal severely compromises the module's function. Despite their importance, these components are not absolutely conserved between parts of the tree of life, suggesting that cells can evolve to perform the same biological functions with different proteins. We evolved *Saccharomyces cerevisiae* for 1000 generations without the important polarity gene *BEM1*. Initially the *bem1Δ* lineages rapidly increase in fitness and then slowly reach >90% of the fitness of their *BEM1* ancestors at the end of the evolution. Sequencing their genomes and monitoring polarization reveals a common evolutionary trajectory, with a fixed sequence of adaptive mutations, each improving cell polarization by inactivating proteins. Our results show that organisms can be evolutionarily robust to physiologically destructive perturbations and suggest that recovery by gene inactivation can lead to rapid divergence in the parts list for cell biologically important functions.

**\*For correspondence:**
l.laan@tudelft.nl

**Present address:**
†Bionanoscience Department, Delft University of Technology, Delft, Netherlands

**Competing interests:** The authors declare that no competing interests exist.

**Reviewing editor**: Yitzhak Pilpel, Weizmann Institute of Science, Israel

## Introduction

Advances in cell biology, genetics, and systems biology have led to substantial understanding of how cells perform complex tasks precisely. In cell polarization and movement, a biochemical and biophysical picture is emerging of how those complex functional modules self-organize to accomplish their functions (*Howard et al., 2011*; *Goehring and Grill, 2013*). Surprisingly, components that are essential for a module in well-studied model organisms can be absent in evolutionarily distant organisms (*Bergmiller et al., 2012*), even though the modules must perform the same tasks. This observation suggests that complex modules reorganize during evolution, either to accommodate changing requirements or to respond to the chance loss of components during population bottlenecks, when selection against deleterious mutations is greatly diminished.

One approach to understanding the evolution of functional modules is to compare them between different species (*Carvalho-Santos et al., 2011*; *Azimzadeh et al., 2012*; *Vleugel et al., 2012*). In closely related, inter-fertile species, genetic analysis can reveal the mutations that account for functional differences, but not their temporal order, and even this level of detail cannot be achieved in more distantly related species. Experimental microbial evolution circumvents these problems: sequencing and genetic analysis identifies the mutations responsible for the selected phenotype and storing and analyzing intermediate steps reveals the order in which mutations occurred (*Lenski and Travisano, 1994*; *Lang et al., 2013*). In principle, these tools should lead to mechanistic understanding of evolutionary trajectories, but selections for faster growth or novel functions typically produce adaptive mutations in multiple functional modules (*Kvitek and Sherlock, 2013*), whose relationship to each other is hard to explain. Are there multiple solutions to the selection, resulting in independent additive solutions in different cellular modules (*Khan et al., 2011*; *Koschwanez et al., 2013*), or are those mutations (and the modules they lay in) coupled in an unknown way (*Wildenberg and Murray, 2014*)?

**eLife digest** Cells use the genetic instructions provided by genes in particular combinations called 'modules' to perform particular jobs. Very different organisms can share many of the same modules because certain abilities are fundamental to the survival of all cells and so they have been retained over the course of evolution. That said, these modules may not necessarily involve the same genes because it is often possible to achieve the same result using different components.

One way to study how those modules can diversify is to deliberately disrupt one of the genes in a module, and observe how the organism and its descendants respond over many generations. Other genes in these organisms may acquire genetic mutations that enable the genes to take on the role of the missing protein. However, the removal of a single component can be detrimental to the survival of the organisms or may affect many different processes. This can make it difficult to understand what is going on.

A gene called *BEM1* is crucial for yeast cells to establish polarity, that is, to allow the different sides of a cell to become distinct from one another. This activity is essential for the yeast to replicate itself. Previous studies have shown that the *BEM1* gene had a different role in other species of fungi, which suggests that yeast may have other genes that previously assumed the role that *BEM1* does now. In this study, Laan et al. removed *BEM1* from yeast and allowed the population of mutant cells to evolve for a thousand generations. The approach differs from previous studies because Laan et al. deliberately selected for yeast that had acquired multiple genetic mutations that can together almost fully compensate for the loss of *BEM1*.

Initially, the mutant cells grew very slowly, were abnormal in shape and likely to burst open. However, by the end of the experiment, the cells were able to grow almost as well as the original yeast cells had before the gene deletion. Genetic analysis revealed that the deletion of *BEM1* triggers the inactivation of other genes that are also involved in the regulation of polarity, which largely restored the ability of the disrupted polarity module to work. This restoration follows a 'reproducible trajectory', as the same genes were switched off in the same order in different populations of yeast that were studied at the same time.

The work is an example of reproducible evolution, whereby a specific order of changes to gene activity repeatedly enables cells with severe defects in important processes to adapt and restore a gene module, using whatever components they have left. The next challenge will be to understand how the particular roles of important modules affect their adaptability.

We focused selective pressure by allowing populations to evolve after deleting an important gene in a well-described module. This approach differs from traditional suppressor screens, which isolate single compensatory mutations, by selecting for combinations of mutations, which together significantly increase fitness. The module we perturbed was polarization in budding yeast (*Smith et al., 2002*; *Slaughter et al., 2009a*; *Howell et al., 2012*; *Freisinger et al., 2013*; *Gong et al., 2013*; *Klunder et al., 2013*; *Wu and Lew, 2013*; *Kuo et al., 2014*). Polarization involves selection of an axis of polarity, followed by the asymmetric organization of cytoskeletal elements and membranous organelles and cell wall growth along this axis. Yeast cells polarize and bud by localizing and activating the small GTPase, Cdc42, at a single site (*Slaughter et al., 2009b*; *Wu and Lew, 2013*). In haploid cells, polarization is directed by a historical mark deposited in the previous cell cycle, but even when the mark is absent, yeast cells still polarize efficiently, albeit at a random location (*Chant and Herskowitz, 1991*). Under these conditions, symmetry breaking depends on at least two pathways: (1) an actin-based mechanism based on the positive feedback between actin-mediated delivery of Cdc42 to the plasma membrane and actin polymerization stimulated by membrane-bound Cdc42 (*Wedlich-Soldner et al., 2003*; *Marco et al., 2007*; *Freisinger et al., 2013*; *Slaughter et al., 2013*), and (2) an actin independent, Turing type mechanism that depends on interactions amongst proteins that regulate the activity and localization of Cdc42 (*Howell et al., 2012*; *Freisinger et al., 2013*; *Smith et al., 2013*) (*Figure 1A*). We strongly perturbed yeast polarization by removing Bem1, a regulator of Cdc42, which recruits the guanine exchange factor (GEF) Cdc24 as well as Cdc42 to the membrane where Cdc24 activates membrane-bound Cdc42. We decided to delete Bem1 because previous research suggests that this protein is a relatively young component in polarity establishment

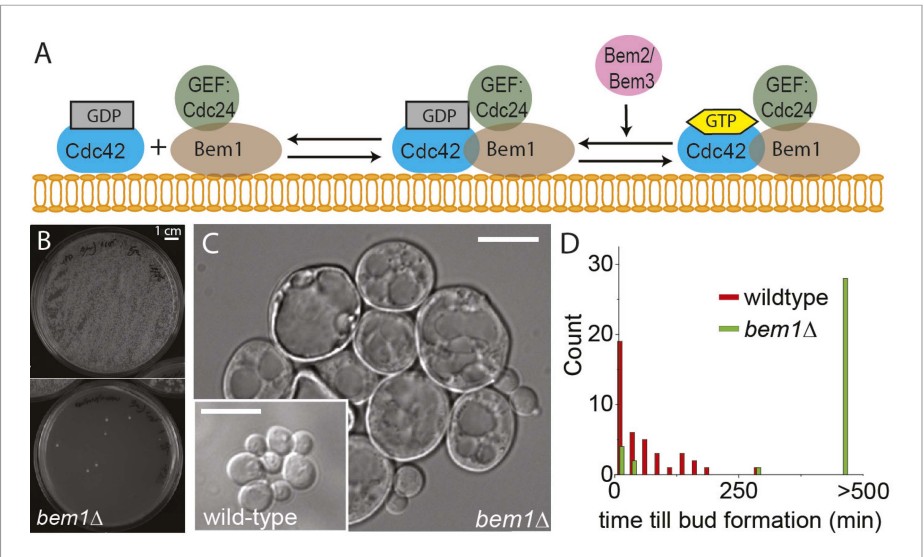

**Figure 1**. *BEM1* is an important polarity gene. (**A**) Cartoon showing the components of the machinery for cell polarization relevant for this work. (**B**) Images of two plates where $10^5$ wild-type (left) vs $10^5$ *bem1Δ* spores were plated. (**C**) Differential interference contrast (DIC) images of *bem1Δ* and wild-type cells that grew from a single spore, showing that *bem1Δ* cells do not polarize effectively and therefore grow very large compared to wild type. Scale bar indicates 20 μm. (**D**) Histograms of the time between cytokinesis and bud formation for *bem1Δ* and wild-type cells that were born after germination from spores.

The following figure supplement is available for figure 1:

**Figure supplement 1**. DIC microscopy image of a microcolony that grew from a single *bem1Δ* cell that was taken from the colony that was used to start the experimental evolution experiment of the A-lines.

that had a different role in ancestral fungi (*Semighini and Harris, 2008*). Deleting this component of polarity could thus reveal other alternative, more evolutionarily ancient polarization modules.

Deleting Bem1 led to profound defects in cell polarization and proliferation, but cells recovered to nearly wild-type growth rates over 1000 generations. Genetic analysis revealed that this recovery followed a reproducible trajectory in which the same genes, which regulate polarization, were inactivated in the same order. Systematic analysis of the interactions between the mutated genes revealed epistatic interactions that explained the evolutionary trajectory that gradually improved cell polarization. We discuss the role of loss-of-function mutations in the evolution of populations outside the laboratory.

## Results

### Rapid evolution in *bem1Δ* lineages

We started by constructing *bem1Δ* cells, in the W303 strain background, by sporulating a heterozygous *BEM1/bem1Δ* diploid. *bem1Δ* spores formed colonies at a frequency of $6 \pm 0.4 \times 10^{-5}$ (*Figure 1B*, for details see Supplementary materials), while wild-type (*BEM1*) spores formed colonies at a frequency of 0.95. We imaged *bem1Δ* and wild-type spores as they germinated and followed several subsequent cell divisions (*Video 1*), where we measured the time of budding as a proxy for polarization: cells cannot bud without a successful polarization event. The majority of *bem1Δ* cells (28/35 vs 0/41 for wild-type cells) did not polarize within 500 min but grew isotropically resulting in very large cells (*Figure 1C*) that often lysed (16/35) (*Video 2*). The *bem1Δ* cells that did polarize, (*Figure 1D*, $P_{polarize\_bem1Δ} = 0.23$, N = 35), polarize fast, in contrast to wild-type cells, which show a wider distribution of polarization times, but always eventually polarize successfully (*Figure 1D*, $P_{polarize\_wt} = 1$, N = 41). We used a single *bem1Δ* colony, as well as a control, wild-type colony, as the starting point for our evolution experiments. From both colonies, we started with 10 wild-type and 10

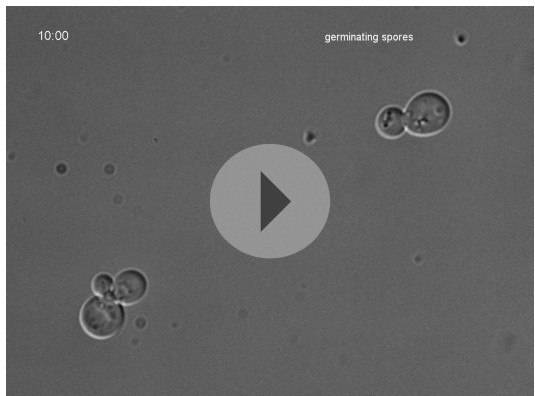

**Video 1.** DIC microscopy video of germinating spores (time step is 3 min) growing in a micro fabricated channel with constant media supply. The spore in the bottom-left is wild-type and the spore in the top-right is *bem1Δ*. The video plays at 2520× real time and the time stamps show hh:mm.

*bem1Δ* cultures, which we evolved for one thousand generations by serial dilutions, regularly freezing down a sample (*Figure 2A* and see Materials and methods), resulting (due to contamination) in nine surviving *bem1Δ* lineages and seven wild-type lineages. We characterized their phenotypic trajectories by measuring the population growth rate at different time points. The initial growth rate of the *bem1Δ* lineages was approximately 12 times lower than the wild-type growth rate. By the end of the experiment, however, the *bem1Δ* cells grew at almost the rate of their wild-type ancestors (*Figure 2B*). Two *bem1Δ* lines, A1 and A2, are plotted individually because they are discussed in more detail later in the article, and one line, A8, became diploid and was excluded from further analysis. We examined how the cell size distribution, which is an approximation for polarization dynamics (*Figure 2—figure supplement 1E*), changed during the experiment: cells that take longer to polarize are on average larger than cells that polarize fast because yeast cells continue to grow during polarity establishment (*Goranov et al., 2009*). We measured cell size distributions (*Figure 2D*, see Materials and methods) and fitted them to a log normal distribution to determine the mode, as a measure of the dominant cell size, and standard deviation, which we take as an approximation for noisiness in polarization dynamics. At the beginning of the experiment, *bem1Δ* cells were larger and showed a wider distribution than the wild-type ancestor (*Figure 2C,E,F*) as confirmed by microscopy (*Figure 1—figure supplement 1*, *Figure 1* and *Figure 2D*). At the end of the experiment, however, both the mode and the standard deviation in *bem1Δ* cells adapted to wild-type cell size (*Figure 2D*) and growth rate values, suggesting a properly functioning polarization machinery.

## Reproducible evolutionary trajectories in parallel lineages

Which mutations caused the changes in growth rate and cell size? We sequenced the whole genomes of several evolved lines: from the seven wild-type lines that were left at the end of the evolution experiment, we sequenced five lines. We sequenced a total of 10 *bem1Δ* lines: the eight *bem1Δ* A-lines that arose from the same starting colony and remained haploid, as described above. In addition, we sequenced two *bem1Δ* lines that were evolved from two independent different starting colonies in a trial experiment (T2 and T3) (*Laan et al., 2015*). In the control, wild-type lines we found a diverse set of mutations (*Supplementary file 1*), with only one gene, *ECM21*, being mutated twice. *ECM21* was also the only mutated gene, in our control lines, that was also found in either of two other large scale evolution experiments (*Lang et al., 2013*; *Kryazhimskiy et al., 2014*), suggesting that adaptation in our control experiments involves similar mutations to those in other studies that have led to modest increases in the proliferation rate of wild-type cells. This is in contrast to the mutations

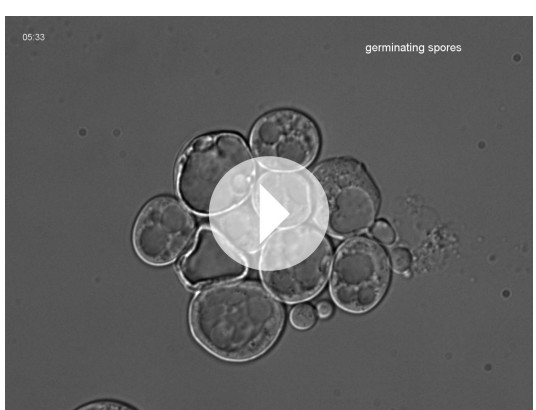

**Video 2.** DIC microscopy video of a microcolony that grew from an individual *bem1Δ* spore (time step is 3 min) in a micro-fabricated channel with constant media supply. Both very large cells, which often lyse, and small, fast dividing cells can be observed. The video plays at 2520× real time and the time stamps show hh:mm.

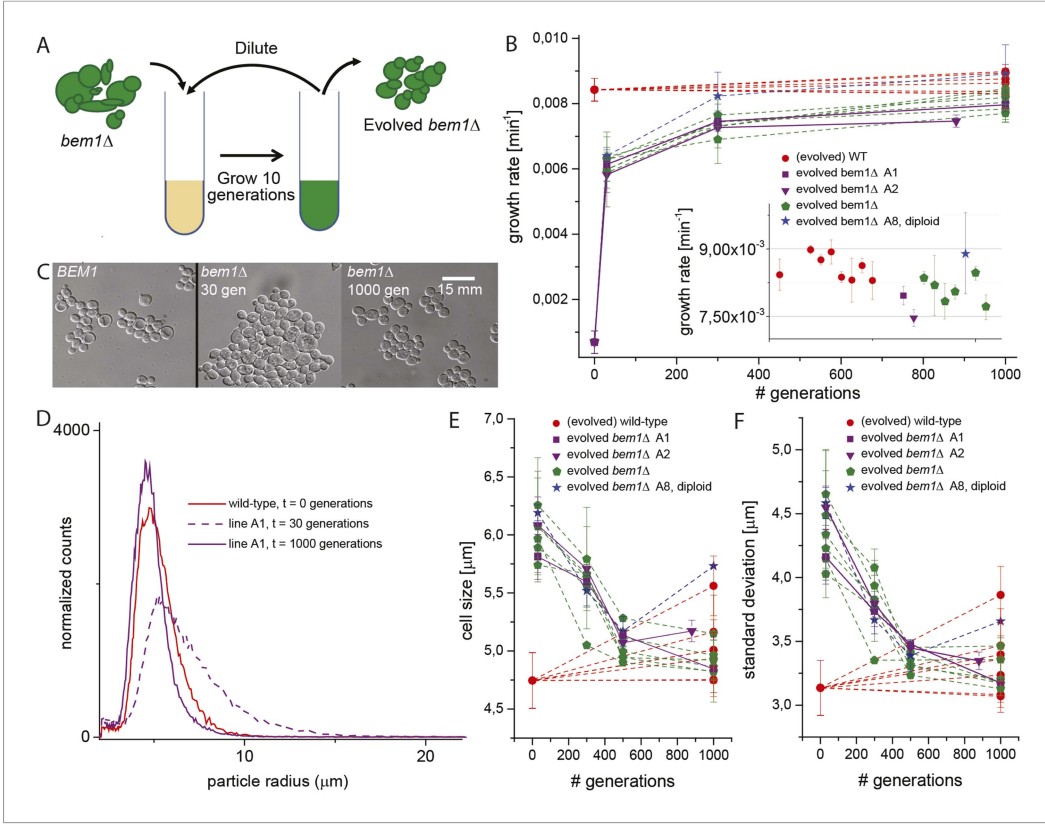

**Figure 2**. Experimental evolution experiments reveal that *bem1Δ* cells can adapt to wild-type growth rate and cell shape in 1000 generations. (**A**) Cartoon of experimental evolution of *bem1Δ* and wild-type for 1000 generations by 100 cycles of serial dilution. (**B**) The log phase growth rate in bulk was measured for several time points (in number of generations) of the evolution experiment, for different *bem1Δ* (A1–10) and wild-type lines. The insert shows that the growth rates of *bem1Δ* cells are close to, but in aggregate significantly lower (t-test, p-value < 1e$^{-5}$) than the growth rate of wild-type cells at the end of the evolution experiment. (**C**) DIC images of wild-type cells (left) and *bem1Δ* cells from line A1 at 30 (middle) and 1000 generations (right), showing changes in cell sizes within and between populations. (**D**) Cell size distribution for wild-type cells (red) and *bem1Δ* cells from line A1 at 30 (dashed, purple) and 1000 generations (solid, purple). These data are fitted to a log normal distribution to find the mode (peak location) and standard deviation. Subsequently, the mode (**E**) and standard deviation (**F**) are plotted for several time points of the evolution experiment for *bem1Δ* (A1–10) and wild-type lines. Note that the mode and standard deviation significantly increased in one of the wild-type lines due to increased clumping (*Figure 2—figure supplement 1*). The error bars indicate SD between independent experiments.

The following figure supplement is available for figure 2:

**Figure supplement 1**. Variable clumpiness in the evolved wild-type lines.

we found in the 10 *bem1Δ* lines. We found three genes that were mutated at least three times: *BEM3* (10/10), *NRP1* (5/10), and *BEM2* (3/10). Mutations in these genes were not reported in a variety of other yeast evolution experiments (*Gresham et al., 2008*; *Kvitek and Sherlock, 2011*; *Koschwanez et al., 2013*; *Kvitek and Sherlock, 2013*; *Lang et al., 2013*; *Kryazhimskiy et al., 2014*), suggesting that they are specific for the deletion of *BEM1*. All other mutations can be found in *Supplementary file 1*. Only one of these mutations, the mutation in *IRA1*, in line T3, is also commonly found in other yeast evolution experiments suggesting that it is not specific for the deletion of *BEM1* (*Figure 3A*). The A-lines shared the same early stop mutation in *BEM3*, but lines T2 and T3 independently acquired different early stop mutations in *BEM3*. The *BEM3* mutation in the A-lines (Q61*) was acquired after germination of the spore that acted as their ancestor: cells from the original colony (*Video 3*) showed the same severe growth defects as freshly germinated *bem1Δ* spores, whereas engineered *bem1Δ*

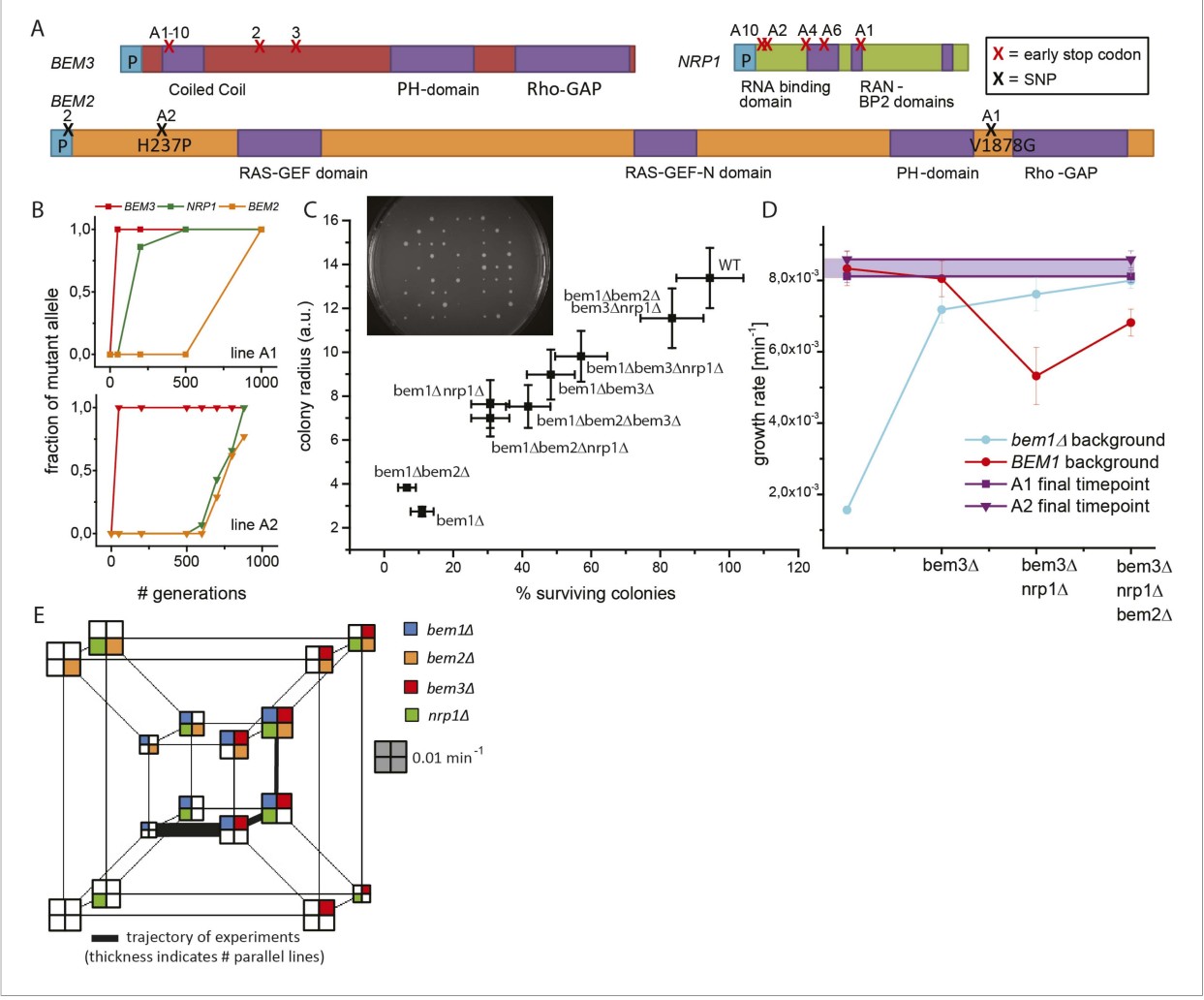

**Figure 3.** Three mutations produce adaptation to deletion of *BEM1*. (**A**) Locations of the mutations in the three genes that were mutated at least three times. The purple color indicates different functional domains in the genes. The three genes and locations of the mutations are drawn roughly to scale. (**B**) For *bem1Δ* line A1 and A2 the fraction of the mutant vs the wild-type allele in the population is plotted against the number of generations in the evolution experiment. (**C**) We sporulated a diploid yeast strain (*BEM1/bem1Δ::KanMx, BEM3/bem3Δ::Nat, NRP1/nrp1Δ::HphMx* and *BEM2/bem2Δ::LEU2*), to obtain all different combinations of mutations. Subsequently, we spotted those spores on plates (insert) and measured for every genotype, the percentage of macroscopic colonies forming spores (the error bar is the statistical error) as well as the average radius of those colonies (the error bar is the standard deviation). (**D**) The log phase growth rate in bulk (the error bar is the standard deviation) is measured for the reconstructed intermediates of *bem1Δ* cells in line A1 and A2 (dark blue). The red line indicates the effect of the three adaptive mutations in a wild-type background. For comparison, the purple dots and line indicate the difference between the growth rates of the evolved lines A1 (faster) and A2 (slower) at the end of the evolution experiment. (**E**) Hypercube where the genotype is depicted by the location and the color in the quadrant. The different paths on the hypercube represent all the different trajectories between any two genotypes within this genotype space. The area of the square represents the growth rate of that genotype, as indicated by the gray scale bar square. The outer-cube represents all *BEM1* lines, which are in our evolutionary experiment inaccessible, because the *BEM1* gene was completely removed from the genome, however, these data are included to reveal the relative change in *BEM1* dependence in different genetic backgrounds.

The following figure supplement is available for figure 3:

**Figure supplement 1.** Reconstructing the evolutionary trajectory.

*bem3Δ* cells had a much less severe defect. All five mutations in *NRP1* (all in A-lines) were independently acquired early stop mutations, whereas in *BEM2* we found a promoter mutation (line T2) and two amino acid substitutions (line A1 and A2), which are radical substitutions at conserved positions.

Lines A1 and A2 acquired mutations in *BEM2*, *BEM3*, and *NRP1*. Sanger sequencing of the three genes at different time points revealed that those mutations occurred sequentially in *BEM3*, *NRP1*,

and *BEM2* in both lines (*Figure 3B*). We investigated whether this order was coincidental or caused by epistasis, by investigating the phenotypes of mutations in these three genes, in different combinations, in a *bem1Δ* background. We approximated the mutations by gene deletions, assuming that the evolved mutations had eliminated (in the case of nonsense mutations) or diminished (amino acid substitutions) gene function. We generated the different genotypes by sporulating a *BEM1/bem1Δ BEM2/bem2Δ BEM3/bem3Δ NRP1/nrp1Δ* diploid (*Figure 3—figure supplement 1A*). For each genotype, we measured the average colony size generated by germinating spores, the percentage of spores that produced visible colonies, and the growth rate in liquid media (*Figure 3E*, *Figure 3—figure supplement 1B,C*). *Figure 3C* shows that *bem1Δ* spores are least likely to form colonies, while *bem1Δ bem2Δ bem3Δ nrp1Δ* spores closely resemble wild-type spores, confirming that inactivating *BEM2*, *BEM3*, and *NRP1* substantially suppresses the severe polarization defect of *bem1Δ* cells. We compared the growth rate of quadruply mutant *bem1Δ bem2Δ bem3Δ nrp1Δ* cells with the final populations of line A1 and A2 (*Figure 3D*). The growth rate of line A2 is indistinguishable from the reconstructed strain, which carries gene deletions in the genes that were mutated in our experiments, suggesting that we found the mutations that confer fitness to this lineage, but line A1 grows slightly faster than the quadruple mutant, suggesting that it may contain additional adaptive mutations.

What does the successful reconstruction tell us? First, it reveals epistasis, in particular, sign epistasis (a mutation switching from being beneficial to deleterious) for some combinations of mutations. If mutations are added in the order they occurred in during the evolution, each successive mutation increases the fitness of the resulting strain, but adding mutations in different orders can reduce fitness. (*Figure 3C*): adding the *bem2Δ* mutation increases the growth rate of *bem1Δ bem3Δ nrp1Δ* cells, but the same mutation reduces or has no effect on the fitness of all other tested genotypes. Another example of epistasis is the changing dependence of the cells on *BEM1*: the polarization module evolves from requiring Bem1, in wild-type strains, to being slightly impaired by the presence of functional Bem1 in a *bem2Δ bem3Δ nrp1Δ* mutant (*Figure 3D*).

## Adaptive mutations alter the dynamics of cell polarization

Reconstruction allowed us to examine how different combinations of mutations alter the dynamics of the polarization module. Because there are large cytoplasmic pools of Cdc42, which make it hard to monitor the location of membrane-associated, active Cdc42, we examined polarization by imaging Spa2-Citrine, which localizes to the polarity site and the cytokinetic ring (*Figure 4A,B*, *Video 4*) (*Snyder, 1989*). We measured two parameters: $t_{fs-c}$, the average time between cytokinesis $t_c$ and the appearance of the first polarity site $t_{fs}$, and how long cells spent with zero, one, two or three polarity sites in the interval between the appearance of the first focus of Spa2-Citrine and budding. As expected, *bem1Δ* cells have a long delay before the first signs of polarization, and polarization is often abortive (*Video 5*, *Figure 4C*), suggesting weaker activation of Cdc42. In addition, *bem1Δ* cells often contain more than one polarity site, suggesting that the positive feedback that ensures a single site of polarization is weaker.

The nature of the mutated genes suggests how inactivating them improves the polarization of *bem1Δ* cells. *BEM3* and *BEM2* encode two of the four GAPs (GTPase Activating Proteins) in the polarity module that inactivate Cdc42 (*Zheng et al., 1994*) and other small G proteins by stimulating their intrinsic GTPase activity. Inactivating these genes should increase Cdc42 activity. As predicted, *bem1Δ bem3Δ* cells bud faster (*Video 6*, *Figure 4C*), but the cells still often contain more than one polarity site, suggesting that increasing Cdc42 activity is not enough to re-establish precise positive feedback in the absence of Bem1.

Less is known about *NRP1*, an RNA binding protein that localizes to stress granules and has not been related to polarity before (*Buchan et al., 2008*). A high-throughput RNA binding study suggested seven target RNAs for *NRP1*, but none of them forms an obvious link to polarity, and most have an unknown function (*Hogan et al., 2008*). The most obvious change after *NRP1* deletion is a decrease in time between cytokinesis and the appearance of the first polarity site (*Video 7*, *Figure 4C*). Therefore, we speculate that *NRP1* plays a role in initiating polarization, which is triggered when Cdc28 is activated by G1 cyclins. Cdc28 activation releases Cdc24 from the nucleus, activating Cdc42. If the absence of Nrp1 leads, directly or indirectly, to the release of more Cdc24, this would lead to faster activation of Cdc42 and earlier polarization. Finally, the loss of Bem2 reduces the time that the cells are not polarized, suggesting that this mutation also increases Cdc42 activity (*Video 8*).

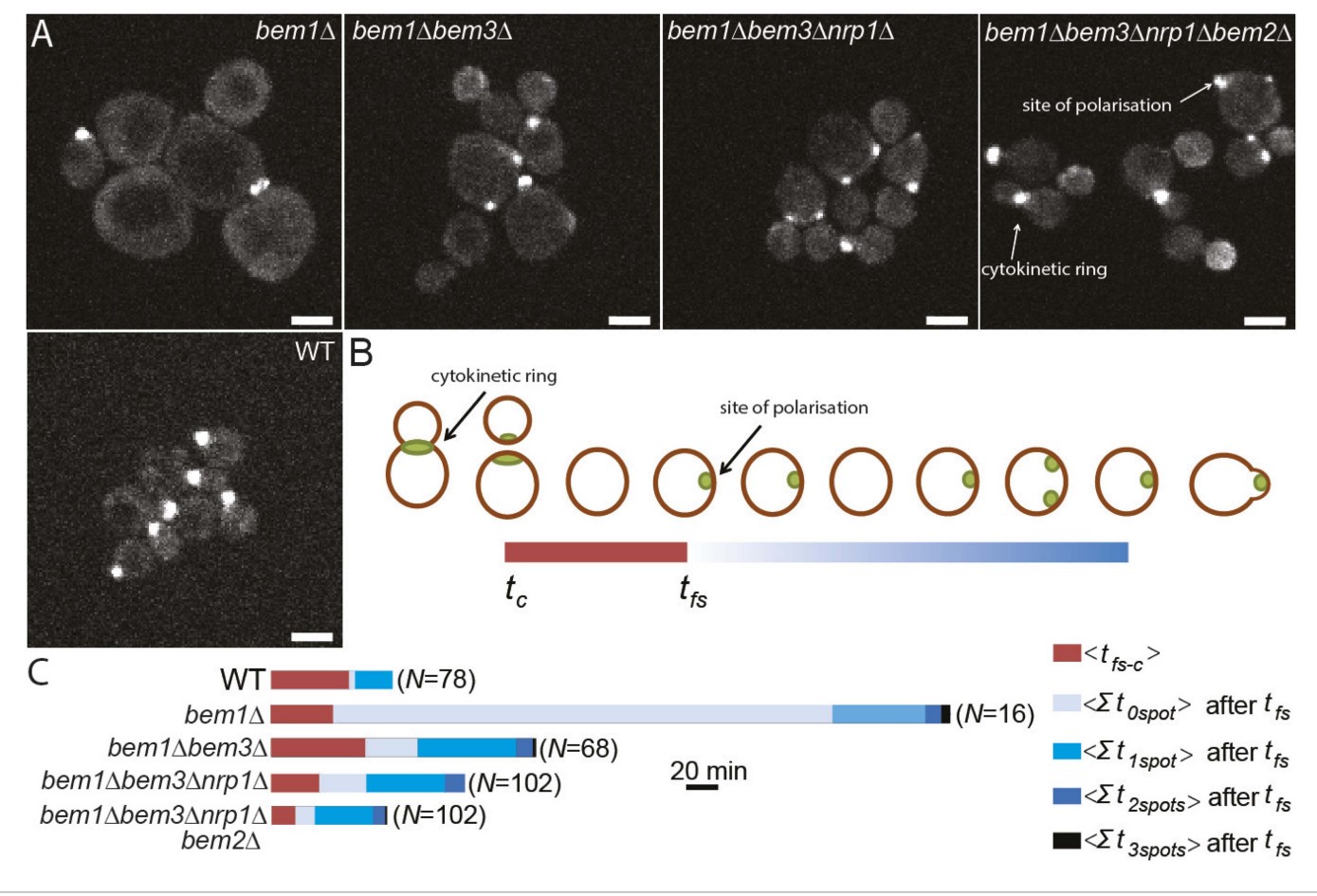

**Figure 4**. Three adaptive mutations change polarization dynamics to re-establish efficient polarization. (**A**) Z-projections of spinning disk fluorescence microscopy images of Spa2-Citrine, which marks the site of polarization as well as the cytokinetic ring, in yeast cells at different reconstructed stages of the evolution. The scale bar indicates 10 μm. (**B**) Cartoon explaining our analysis of polarization dynamics. For the indicated number of mother cells (*N*), we average (1) the time ($t_{fs-c}$) between cytokinesis ($t_c$) and the moment that the first polarity spot appears ($t_{fs}$), indicated in red, and (2) the total time per cell cycle, after $t_{fs}$, when cells contained zero, one, two, or three polarity spots (for details see Supplementary materials). The different times are combined in a horizontal bar plot (**C**) where the length of the bars indicates the average time in minutes.

Our data suggest that that multiple mutations can restore Cdc42 activity without restoring Bem1's ability to physically connect Cdc24 and Cdc42. As a consequence, actin-based positive feedback, which only requires Bem1 to activate Cdc42, will become critical because actin-independent polarization depends directly on Bem1 acting as a scaffold. Previous work showed that when actin-based positive feedback acts alone, it creates multiple simultaneous polarity sites (*Freisinger et al., 2013*), as we observe in our mutants.

## Discussion

We followed the evolutionary adaptation to the removal of an important component of the module that polarizes budding yeast cells. Cells recovered by following a reproducible evolutionary trajectory that could be largely explained by the interactions amongst the mutated genes and led to a substantial recovery in the speed and accuracy of cell polarization.

How repeatable is evolution? Experimental evolution has produced a range of answers to this question, from replicate populations that share a small fraction of causal mutations (*Gresham et al., 2008*; *Kvitek and Sherlock, 2011*; *Chou and Marx, 2012*; *Koschwanez et al., 2013*; *Kryazhimskiy et al., 2014*) to those where mutations occur in the same genes in the same order (*Blount et al., 2012*;

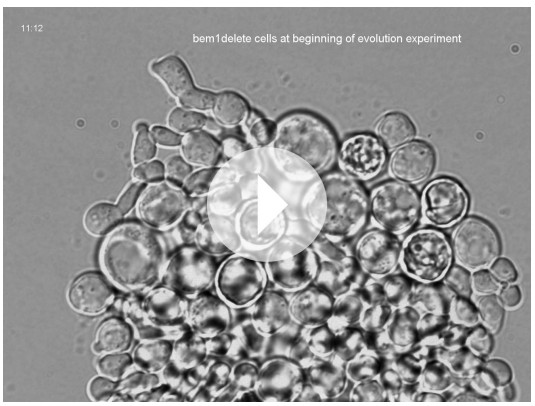

**Video 3.** DIC microscopy video of a microcolony that grew from an individual *bem1Δ* cell, taken from the same colony that was used for the large evolution experiment (A lines). The colony was grown on an agar pad to provide constant culturing conditions (time step is 2 min). The video plays at 2520× real time and the time stamps show hh:mm.

*Herron and Doebeli, 2013*) or repeatedly affect the same amino acids in a protein (*Meyer et al., 2012*). For multiple substitutions, in the same protein, strong epistasis can render most trajectories impossible or unlikely (*Weinreich et al., 2006*; *Bridgham et al., 2009*; *Hinkley et al., 2011*). Examining the mutations that gave rise to a particular phenotype in the bacterium, *Pseudomonas fluorescens*, revealed a hierarchy of pathways: loss-of-function mutations are most frequent and other pathways, including promoter mutations, gene fusions, and gain-of-function mutations can only be found when the dominant pathway is blocked (*Lind et al., 2015*).

Our work adds to the list of examples of reproducible trajectories and our reconstruction of all possible paths, an exercise that has previously been performed for multiple mutations in a single gene (*Weinreich et al., 2006*), demonstrates why our mutations occurred in a particular order. As far as we know, this is the first example of reproducible trajectories to adaptation to mutations that severely compromise a cellular function, and it will be interesting to see if evolving strains, adapting to severe perturbations in other pathways will also lead to reproducible trajectories.

We interpret our results as demonstrating that the multiple layers of regulation that allow cells to polarize rapidly and accurately also make it possible for them to adapt, by evolution, to very strong perturbations. Under strong selective pressure, the larger target size for mutations that inactivate proteins favors evolutionary trajectories that remove components from modules over those that quantitatively alter their properties. As long as modules contain components, like Bem2 and Bem3, which alter the quantitative behavior of other proteins, like Cdc42 and other small GTPases, removing the regulators is the equivalent of rarer mutations that would change the biochemical parameters of central components of the pathway. Our findings are consistent with previous experimental evolution work which showed that loss-of-function mutations allow bacteria (*Zinser et al., 2003*; *Hottes et al., 2013*) and yeast (*Koschwanez et al., 2013*) to adapt to their environments and produce novel phenotypes (*Wildenberg and Murray, 2014*). Adaptation by loss-of-function mutations has also been observed in natural populations (*Maurelli et al., 1998*; *Smith et al., 2006*; *Bliven and Maurelli, 2012*; *D'Souza et al., 2014*, *2015*).

In our work, the selected mutations do not alter the structure of an existing protein to allow it to play Bem1's role of physically linking Cdc42 to Cdc24. Instead, other mutations that increase the half-life of Cdc42-GTP (and possibly other small G proteins) remove the need to hold Cdc24 and Cdc42 close to each other. We suspect that the differing effects of removing genes and altering gene dosage may control the type of mutations that allow cells to adapt to large genetic perturbations. If increasing the dosage of genes can reverse the effects of the perturbation, cells may recover by becoming aneuploid, as observed when duplicating chromosome XVI increases the dosage of two genes (*MKK2* and *RLM1*) that help compensate for the absence of type II myosin (*Rancati et al., 2008*). In contrast, if removing inhibitors will increase fitness, point mutations that inactivate the inhibitors will be selected, as we observed. Investigating how cells recover from other perturbations in these and different pathways will test the validity of this speculation, reveal the mechanistic details underlying evolutionary change, and improve our understanding of how the self-organizing properties of modules affect the course of evolution.

Our results may help to explain the surprising observation that certain eukaryotic lineages lack extremely well-conserved pathways, such as the absence of the anaphase promoting complex from *Giardia* (*Gourguechon et al., 2013*) or conserved kinetochore proteins from kinetoplasids (*Akiyoshi and Gull, 2014*). Although less in known about the evolutionary history of polarity proteins, recent studies suggest that the components of this pathway also vary substantially. In filamentous fungi,

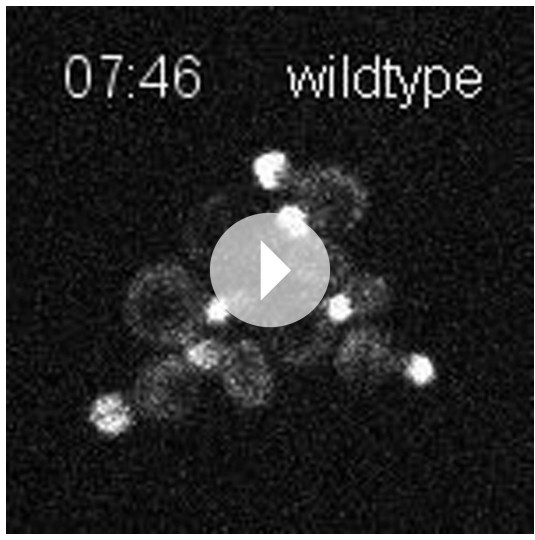

**Video 4.** Spinning disk confocal fluorescence micros-copy video of Spa2-Citrine in wild-type cells. For every image, the cells were exposed for 200 ms with a 488 laser (the time step is 2 min). The video shows a time series of maximum projections of seven z-stacks (z-step is 1 μm). For *Videos 4–8*, the image size and the intensity values are kept constant. The cells are grown in a micro-fabricated channel with constant media supply. As cells divide, Spa2 concentrates at the bud neck and the single fluorescent focus splits into two with both mother and daughter cells inheriting localized Spa2. In mothers this spot often becomes the site where buds emerge, but in daughters, who spend longer in G1, the spot typically disappears and later reappears, at a different site, just before the next bud emerges. The video plays at 2520× real time and the time stamps show hh:mm.

Cdc42 is not essential because its function is shared with Rac1 (*Kwon et al., 2011*). However, the deployment of the Cdc42 module and the Rac1 module for critical morphogenetic functions is surprisingly different between closely related species (*Harris, 2011*). Our study shows that removing one component of a conserved path-way selects for the inactivation of others. If this process was accompanied by the recruitment of novel proteins, it could ultimately lead to an evolutionary shift that replaced one module with another unrelated one that performed the same function.

One objection to this hypothesis is that it would be very hard to fix a mutation as deleterious as *bem1Δ*. There are ways of coun-tering the objection: severely defective muta-tions could be fixed in population bottlenecks that accompany speciation; secondly, and maybe more likely, mutations that removed important components could be pleiotropic, offering advantages in novel environments that were similar in magnitude to the costs they impose on strongly conserved core functions; and thirdly, previously selected mutations in the same or related pathways could make the defects associ-ated with the removal of a component less severe.

Because of the difficulty of inferring events as ancient as the ones that rearranged the kineto-chore of kinetoplastids or made the anaphase promoting complex dispensable in *Giardia*, it is impossible to say whether it was the loss of genes that inactivated existing pathways, or some other event that triggered the sequence of changes that led particular lineages to use different proteins to perform functions that appeared early in eukaryotic evolution. Despite our inability to reconstruct these processes, there is evidence for individual steps in the process of functional reorganization. These include the loss of widely conserved genes in individual evolutionary lineages (*Bergmiller et al., 2012*; *Gourguechon et al., 2013*; *Drinnenberg et al., 2014*), the loss of genes present in ancestral species during evolutionary diversification (e.g., the loss of 88 genes in the descent of *Saccharomyces cerevisiae* from an ancestor that existed 100 Mya ago [*Gordon et al., 2009*]), and the recruitment of new functions by adaptations that alter the function of existing genes and create genes de novo (*Neme and Tautz, 2013*, *2014*).

Our evolution experiment created a related set of rapid polarizing mechanisms. Deciphering the physical mechanisms of polarity establishment in all the different combinations of mutants will teach us about evolution of functional modules. However, it will also reveal more about cell biology of polarization. First, it can help to identify the role of new genes in polarity establishment; previous work has implicated Nrp1 in RNA binding, ribosome biogenesis, and the formation of stress granules, whereas our experiments demonstrate that it regulates cell polarization, directly or indirectly. High-throughput studies, at least as indicated on the yeast genome website SGD (www.yeastgenome.org) have not shown physical or genetic interactions between *NRP1* and any of the currently known polarity genes. *BEM1*, *BEM2*, and *BEM3* as well as their genetic interactions have been measured and implicated in polarity establishment before (*Koh et al., 2010*). However, the lack of information about *NRP1* made it impossible to predict the positive effect of the *BEM2* deletion on polarity establishment in the absence of *BEM1* and *BEM3*.

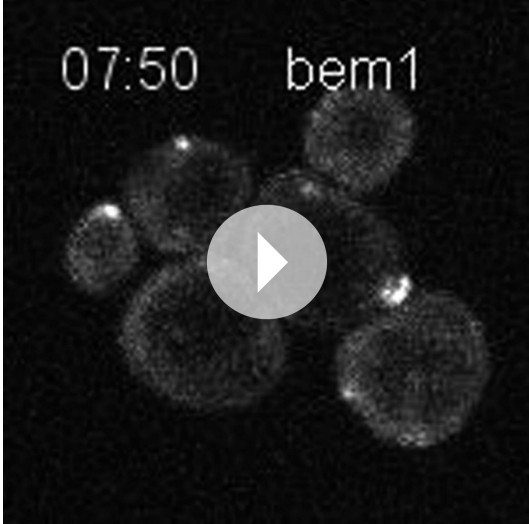
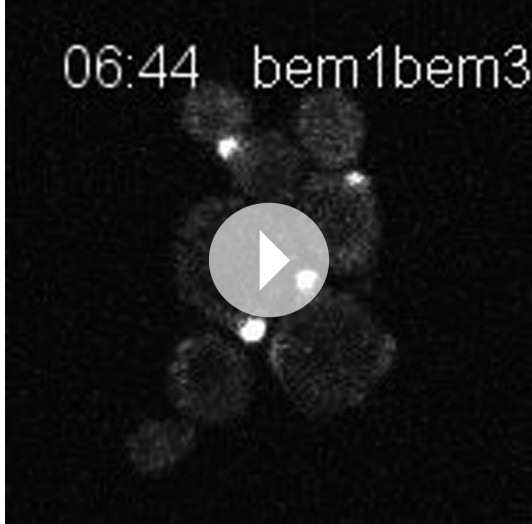

**Video 5.** Cell polarization in *bem1Δ* cells. Conditions are identical to *Video 4*; however, the cells in the video are *bem1Δ*. Note the much slower cell cycles, the appearance and disappearance of Spa2 spots that do not give rise to budding, including one site of Spa2 accumulation that persists for most of the video, but does not lead to a bud emerging. Cells rarely show two simultaneous strong Spa2 spots. The video plays at 2520× real time and the time stamps show hh:mm.

**Video 6.** Cell polarization in *bem1Δ bem3Δ* cells. Conditions are identical to *Video 4*; however, the cells in the video are *bem1Δ bem3Δ* double mutants. Note the frequent presence of two competing spots of localized Spa2. The video plays at 2520× real time and the time stamps show hh:mm.

Second, by biophysically investigating how cells recover from the deletion of *BEM*1 as well as other perturbations in the polarity pathway, we will be able to distill which parts of polarization mechanisms, rather than molecules, are essential for proper function. Cell polarization is a complicated, dynamical process, thus it may be more important to alter quantitative parameters of the overall process, regardless of the gene whose mutation produces the change, than it is to alter the behavior of a particular protein.

## Materials and methods

### Yeast strains/media
The W303 strain background was used for all experiments. *Supplementary file 2* provides a detailed list of each strain used. Standard rich media, YPD (2% Peptone, 2% D–Glucose, and 1% Yeast-Extract) was used for the evolution experiments. For the microscopy experiments non-fluorescent yeast media was used, which was prepared from refrigerated 10× yeast nitrogen base (YNB), 20% D-Glucose (10×), and sterilized water. The amino acids leucine, histidine, and uracil were added from a 100× stock. The YNB was based on the recipe of *Wickerham (1952)*, with the following modifications: riboflavin and folic acid were not added to the YNB to minimize auto fluorescence (*Sheff and Thorn, 2004*). All other media was prepared according to (*Sherman et al., 1978*).

### Generating the various mutant haploid cells
The various haploid strains in our study were generated by sporulating heterozygous diploids (*Figure 1B*, *Figure 3C*). This approach allowed us to select and monitor barely viable mutants from the moment they were created, and allowed us to observe the occurrence of the first and subsequent suppressor mutations. Diploid strains were sporulated in liquid culture by growing them to saturation in YPD at 30°C. Afterwards they were diluted into YEP (2% peptone and 1% yeast-extract) with 2% potassium acetate for 12 hr at 30°C, washed with water, resuspended in 2% potassium acetate and grown at 25°C for 3–5 days. The sporulation efficiency was checked under the microscope. If the sporulation efficiency was high enough (>95%) 1.5 ml of spores were spun down, resuspended in

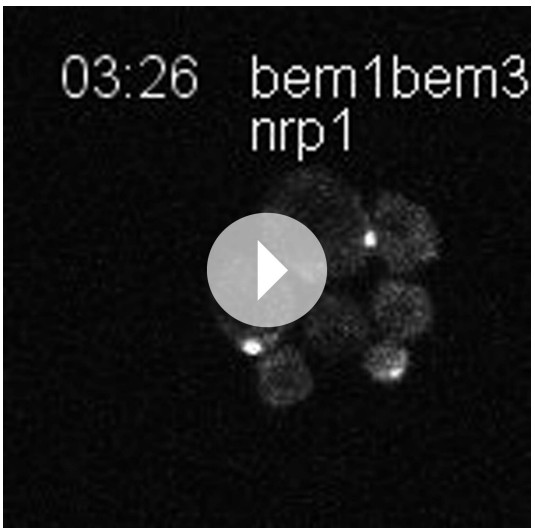

**Video 7.** Cell polarization in *bem1Δ bem3Δ nrp1Δ* cells. Conditions are identical to **Video 4**; however, the cells in the video are *bem1Δ bem3Δ nrp1Δ* triple mutants. Competition between competing Spa2 spots is resolved more quickly than in *bem1Δ bem3Δ* cells. The video plays at 2520× real time and the time stamps show hh:mm.

**Video 8.** Cell polarization in bem1Δ bem3Δ nrp1Δ bem2Δ cells. Conditions are identical to **Video 4**; however, the cells in the video are *bem1Δ bem2Δ bem3Δ nrp1Δ* quadruple mutants. Although these cells bud at similar sizes and rates as *BEM1* cells, some cells show prolonged presence of two or more Spa2 spots and mothers rarely produce their next bud close to the site of its predecessor. The video plays at 2520× real time and the time stamps show hh:mm.

500 µl Zymolyase solution (10 µg/ml Zymolyase in TE (10 mM Tris, 1 mM EDTA, pH 7.5)) and incubated at 36°C for 60 min, to digest the ascus. Afterwards, the spores were spun down, resuspended in 500 µl 0.1% SDS +0.1% Triton in TE and incubated at 36°C for 30 min, to disrupt the membranes of remnant diploid cells. Subsequently, the spores were vigorously vortexed and the spore concentration was measured with a Coulter Counter (Z2 analyser; Beckman Coulter, Inc., Danvers, MA). $10^1$, $10^2$, and $10^3$ spores were plated on YPD plates (3× per condition), to get an estimate of the total number of viable spores. To select for viable *bem1Δ* cells (*Figure 1*), $10^3$, $10^4$, $10^5$, and $10^6$ spores were also plated on 4xG418/10xCANAVINE/-HIS plates (3× per condition), which we will refer to as the selection plates. The selection plate strongly selected for *MAT**a**-haploid spores (*Pan et al., 2004*) that lack *BEM1*: the absence of histidine selects for haploid *MAT**a** strains, because the *MFA1* promoter is expressed only in **a** cells, the presence of canavanine is a second selection for haploids, since the *CAN1* gene dominantly confers sensitivity to canavanine, an arginine analog, and G418 selects for *bem1Δ* because the gene deletion is marked with the bacterial kanamycin resistance gene. After 2 days, the colonies on the YPD plates were counted and used to normalize the number of colonies on the selection plate to calculate the survival probability. The error is the standard deviation between five independent experiments and includes the statistical plating error.

## Characterizing *bem1Δ* mutants

The colonies that grew on the selection plates were checked for the absence of *BEM1* by PCR (later this was confirmed by whole genome sequencing) (*Figure 1C,D*). Typically, ~20% of the colonies were indeed *bem1Δ*. The other cells were most likely haploid, *MAT**a**, cells, aneuploid for chromosome II and thus contained both *BEM1* and the selection marker. Cells from a confirmed *bem1Δ* colony were used to start the evolution experiments. In addition, some cells from the same colony were imaged every 3 min for several hours with a Ti-E inverted microscope (Nikon, www.nikoninstruments.com), with a Perfect Focus System and a 60× DIC oil objective (*Figure 1—figure supplement 1*) while being confined in an agar pad. Germinating *bem1Δ*, as well as wild-type spores were imaged every 3 min for 20 hr in the same microscope but in a microfluidic chamber (CellAsic, Millipore, www.emdmillipore.com)

to allow for constant media supply. The time between cytokinesis and bud formation for mothers was manually determined. Only mothers were analyzed to minimize the effect of cell size on the time between cytokinesis and bud formation.

### Experimental evolution experiments

The evolution experiments were initiated with 10 *bem1Δ* (A-lines) and 10 wild-type cultures (3aA-lines) (*Figure 2A*). The 10 *bem1Δ* cultures were derived from the same starting colony, and the 10 wild-type colonies were derived from an individual colony from the yLL3a strain. The individual colonies were dissolved into 1 ml YPD media and counted. Every independent 10 ml YPD culture was inoculated with $10^6$ cells. The glass tubes were placed in a roller drum at 30°C. On the vast majority of days, we checked the culture density in the morning (10 am $\pm 1$ hr). If the culture density was $<5.10^7$ cell/ml, the cells were grown for another 24 hr, otherwise it was passaged as follows. First, 10 µl of the culture was pipetted into 10 ml of fresh YPD and placed in a roller drum at 30°C. Second, 1 ml of culture was mixed with 500 µl of 50% Glycerol in water and frozen at −80°C. Initially, cells were frozen down after every passage, however after passage 20 this was reduced to every five passages. Early in the evolution experiment 1 *bem1Δ* culture and three wild-type cultures were lost due to bacterial contamination. The other cultures were all evolved for 1000 generations (100 passages), except for line A2 which got contaminated after 33 passages, but was restarted from the frozen stock at passage 30 and was evolved for 820 generation (82 passages).

As a pilot experiment, two *bem1Δ* cultures (T2 and T3) were evolved for 1000 generations according to the same protocol, with the exception that they were taken from independent colonies.

### Growth rate measurements and analysis

The population growth rate was measured at different time points during the evolution experiments (*Figure 2B*). Approximately, $10^5$ cells were taken from the frozen stock and inoculated in 2 ml of YPD. We used YPD from the same batch for the complete evolution experiment as well as for the analysis of the evolution experiment. The cultures were incubated on a roller drum at 30°C overnight, and the next morning, the growth rate in log phase was measured by taking several time points with a Coulter Counter and fitting the data to an exponential function with a home written program in Matlab. For every data point, at least three independent experiments were used. The error bar indicates the standard deviation between the different experiments.

### Cell size measurements and analysis

In addition to the growth rate, the population cell size distribution was measured at different time points during the evolution (*Figure 3D,E,F*). Approximately, $10^5$ cells were inoculated from the frozen stock into 2–5 ml of YPD; for each condition, three independent cultures were started in parallel. The cells were grown overnight in a roller drum at 30°C to reach log phase in the morning. If the cell density was $>4 \times 10^6$ and $<3 \times 10^7$, a sample 50 µl of cells was diluted into 20 ml of Coulter Counter isotone solution. Subsequently, the cuvettes were sonicated on ice for 20 s, to reduce clumpiness. Afterwards 100,000 particles were measured with the Beckman Coulter Multisizer 3. For most conditions, three independent experiments were performed (typically consisting of three independent cultures). The distribution of cells sizes measured was imported in Matlab and fitted with a log-normal distribution to obtain the mode and the standard deviation. The error bars are the standard deviations between different cultures.

### Whole genome sequencing and analysis

Genomic DNA library preparation was performed as in *Wildenberg and Murray (2014)* with an Illumina Truseq DNA kit on a Illumina Hiseq 2000 with 150 base paired end reads (*Supplementary file 1*). The Burrows-Wheeler Aligner (bio-bwa.sourceforge.net) (*Li and Durbin, 2009*) was used to map DNA sequences to the *S. cerevisiae* reference genome r64, which was downloaded from *Saccharomyces* Genome Database (www.yeastgenome.org). The samtools software package (samtools.sourceforge.net) was then used to sort and index the mapped reads into a BAM file. GATK (www.broadinstitute.org/gatk) (McKenna et al., 2010) was used to realign local indels, and Varscan (varscan.sourceforge.net) (*Koboldt et al., 2012*) was used to call variants. Mutations were found using a custom pipeline written in Python (www.python.org) using the Biopython (biopython.org) and pysam (github.com/pysam-developers/pysam) modules. The pipeline (github.com/koschwanez/mutantanalysis) (*Koschwanez et al., 2013*)

compares variants between the reference strain, the ancestor strain, and the evolved strains. A variant that occurs between the ancestor and an evolved strain is labeled as a mutation if it either (1) causes a non-synonymous substitution in a coding sequence or (2) occurs in a promoter region, defined as 500 bp upstream of the coding sequence.

## Analysis of allele frequency with sanger sequencing

Commercial Sanger sequencing returns trace plots, chromatograms that indicate the relative frequency of each base at each position in the sequenced DNA (*Figure 3B*). Trace plots were used to estimate the fraction of mutant alleles in a population at different time points during the evolution. The fraction of mutant alleles in the population was assumed to be the height of the mutant allele peak divided by the height of the mutant allele peak plus the ancestor allele peak. In the time course analysis, values below the approximate background level were assumed to be zero, and values above 95% were assumed to be 100% (*Sherman et al., 1978*). In line A2, the dynamics of the mutant *bem2* and mutant *nrp1* allele are close in time. However, two observations strongly indicate that the *bem2* mutation occurred after the *nrp1* mutation: (1) there is a consistent difference at different time points between the fraction of *nrp1* and *bem2* mutant alleles and (2) the difference between the fraction of the population that contain the *nrp1* and *bem2* mutant alleles at the final time point is confirmed by whole genome sequencing.

## Generation and analysis of reconstructed strains

The various strains carrying gene deletions in the genes that were mutated in our experiments were generated by sporulation of a heterozygous diploid (yLL112a and yLL135a) (*Figure 3C,D*, figure supplement 3BC). The phenotype was determined by replica plating colonies that grew from single spores to all the different plates necessary to detect the different markers (figure supplement 3BC). Single spore colonies were generated by spotting individual spores with a FACS cell sorter (MoFlo Legacy, Dako Cytomation/Beckman Coulter) on YPD plates. 81 spores were spotted per plate on a total of 10 plates. After 2 days, the plates were imaged to measure the colony size. Calibration experiments showed that in this period the colony size is a good approximation for the growth rate of the cells in the colony. Subsequently, the colonies were replica plated to various drop-out and drug plates to determine their genotype. Home written software in Matlab was used to determine the colony size of every colony and to combine it with its genotype. From this data, the average colony size per genotype and the standard deviation between the colony sizes was calculated. In addition, the percentage of surviving colonies was calculated by dividing the number of observed colonies by the number of expected colonies*100%. The error bar is the statistical error (% surviving colonies/$\sqrt{N}$, N is number of observed colonies). Comparing the wild-type data from assaying the behavior of the spores spotted on plates with results obtained from tetrad dissections of wild-type diploids, confirmed that the spore plotting assay did not introduce bias in the growth rates or fraction of surviving colonies. The percentage of surviving *bem1*Δ colonies was higher in this assay than in the original assay (*Figure 1A,B*), which we attribute to the presence of Spa2-Citrine: all the *bem1*Δ colonies that survived contained Spa2-Citrine, even though this allele was heterozygous in the diploid that they were derived from. After replica plating, approximately 105 cells/genotype were inoculated into 2 ml YPD and incubated on 30°C on a roller drum overnight and the next morning the growth rate in log phase was measured by taking several time points with a coulter counter and fitting the data to an exponential function with a home written program in Matlab. For every data point, typically three independent experiments were used. The error bar indicates the standard deviation between different experiments.

## Fluorescence microscopy of reconstructed strains

The polarization dynamics of the reconstructed strains were measured by imaging Spa_Citrine, a polarity marker, present in the reconstructed strains (*Figure 4A*). Cells were grown to log phase and flowed into a microfluidic culture chamber, which allowed for constant culturing conditions (CellAsic, Millipore). The cells were maintained in log-phase by constantly supplying non-fluorescent growth media. Images were taken with a 60× objective on a Nikon inverted Ti-E microscope with a Yokagawa spinning disc unit and an EM-CCD camera (Hamamatsu ImagEM); Citrine was excited with a 488-nm laser; exposure times were 200 ms with a time interval of 2 or 3 min. Typically, 16 positions were imaged for 8 hr per experiment. At each location, a z-stack was taken with seven z-steps of 1 μm. For analysis, a video was created from maximum projections of these z-stacks. For every mutant

(*Figure 4*), two independent experiments (consisting of at least four different locations) were analyzed.

## Data analysis of fluorescence microscopy of reconstructed strains

For every mother cell between cytokinesis and bud take off, we manually determined, at each frame, how many Spa2 spots were present (*Figure 4C*). Only mothers were considered to minimize the effect of cell size on the data. The time between cytokinesis and the first spot and the time between the first spot and bud appearance were analyzed separately. From this data, the average total time per cell cycle that cells contain zero, one, two, or three polarity spots was calculated.

## Acknowledgements

We thank the Boeke lab for providing us with the magic marker, Gregg Wildenberg for help with cell sorting, Miguel Coelho and Phoebe Hsieh for help with library generation, Bertus Beaumont, Michael Desai, Vlad Denic, Cassandra Extavour, Michael Laub, and Melanie Mueller for critical reading of the manuscript, Roland Wedlich-Soldner and members of the Murray and Nelson labs for useful discussions. This work was supported by grant GM06783 from the National Institute of General Medical Sciences. LL gratefully acknowledges support from the Netherlands Organization for Scientific Research through a Rubicon grant, as well as from the Human Frontiers Science Program through a cross disciplinary post-doctoral grant.

## Additional information

### Funding

| Funder | Grant reference | Author |
|---|---|---|
| Nederlandse Organisatie voor Wetenschappelijk Onderzoek (NWO) | | Liedewij Laan |
| Human Frontier Science Program (HFSP) | LT000739-2010-C | Liedewij Laan |
| National Institute of General Medical Sciences (NIGMS) | GM06783 | Andrew W Murray |

The funders had no role in study design, data collection and interpretation, or the decision to submit the work for publication.

### Author contributions

LL, Designed the research, Performed the research and analysed the data, Wrote the paper; JHK, Performed whole genome sequence data analysis; AWM, Designed the research, Wrote the paper

## Additional files

### Supplementary files

• Supplementary file 1. Mutations in the evolved lines.

• Supplementary file 2. Yeast strains.

### Major dataset

The following dataset was generated:

| Author(s) | Year | Dataset title | Dataset ID and/or URL | Database, license, and accessibility information |
|---|---|---|---|---|
| Laan L, Koschwanez JH, Murray AW | 2015 | Sequences from Evolutionary adaptation after crippling cell polarization follows reproducible trajectories | http://www.ebi.ac.uk/ena/data/view/PRJEB11363 | Publicly available at the EBI European Nucleotide Archive (Accession no: PRJEB11363). |

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
