## [Decision Letter]

Thank you for submitting your work entitled “Evolutionary adaptation after crippling cell polarization follows reproducible trajectories” for peer review at *eLife*. Your submission has been favorably evaluated by Naama Barkai (Senior editor), Yitzhak Pilpel (guest Reviewing editor), and two reviewers. While there is a consensus that the work is intriguing, and done and written properly, some revision is requested. In particular, as you will see below, the reviewers would like you to better explain the relevance of your study to natural evolutionary scenarios.

Summary:

All referees and the editors agree that the work is exciting. The reproducibility of the evolutionary adaptations, the identity of the mutated genes, and more strikingly, the order of their appearance, are certainly very interesting, especially since a sound epistatic basis for such order is provided.

Yet, both referees questioned the relevance to natural evolution. As clearly put by referee #1: “And it is not even clear whether a Bem1 deletion strain would ever get the chance to evolve in a natural setting as it might be immediately out competed by neighboring wild type cells” and referee #2: “does this starting point, appearance of crippling mutation, reflect the starting point of natural evolutionary scenarios as we understand them?”

Essential revisions:

Especially as suggested by referee #2 a much improved discussion should ground the work in natural evolution. Even when passing through very narrow bottlenecks, how likely it is that mutants that experience gene inactivation would be isolated from competition for 1000 generations in a way that would allow them to reveal evolutionary dynamics of the type found here?

Points 1-3 in referee #2's report are highly essential and the authors are thus asked to address them each in the revised Discussion.

Reviewer #1:

Laan et al. present a study of what happens to yeast cells after BEM1, a gene that is critical for maintaining cell polarity, is deleted. The authors grow the cells for 1000 generations and find that, in several independent lines, the cells evolve to compensate for the loss of BEM1. The striking result is that in several independent lines the same three genes are mutated in the same order during the course of the growth. Through the creation of the double and triple mutants the authors show that the order in which the mutations are acquired is likely due to a complex set of epistatic interactions among the three genes (BEM2, BEM3, and NRD1). The authors conclude that cells can overcome inactivating mutations in critical genes and that gene inactivation may be a mechanism that leads to rapid divergence in the component genes required for particular cellular functions.

Overall all the experiments are well performed, the results are interesting, and the data generally support the conclusions. Similar to other evolutionary studies these experiments represent a specific case and it is important not to generalize the results too much. Yeast may not respond to other deletions in the same way. And it is not even clear whether a Bem1 deletion strain would ever get the chance to evolve in a natural setting as it might be immediately out competed by neighboring wild type cells. Nevertheless, this study represents an interesting case study. I only have a few minor comments:

1) Evolution by gene loss has been written about extensively and the authors should cite some of the previous literature in this area.

2) It was not easy to follow exactly how many independent lines were studied. The text says nine mutant lines were evolved, but then ten lines were sequenced. The authors should clarify exactly how many independent lines there were, where they came from, and discuss in more detail which mutations arose independently and which are shared because of common ancestry.

Reviewer #2:

The paper is well written and clear. It compellingly outlines the adaptation dynamics of a multi-step evolutionary process after the deletion of a major component of the yeast polarization module. The authors used various complementary approaches, ranging from classical genetics to microscopy and whole genome sequencing, to interpret their findings and support their main conclusions. However, my main concern is that the experimental system the authors have devised, the recovery of a lineage from deletion of a crucial component, is highly contrived and does not shed new light on the evolutionary process in natural habitats. Thus I think that one of the main questions the authors wish to address – how do functional modules reorganize during evolution, is not adequately answered.

1) While most experimental evolution works focus on adaptation of lineages to new environments or selective challenges, this study sets off by crippling a strain through mutation of an almost essential gene. The authors rightfully state that this mutation will necessitate reorganization of the functional module or take-over of a new module. Yet does this starting point, appearance of crippling mutation, reflect the starting point of natural evolutionary scenarios as we understand them? The authors only state that such module reorganization may occur to accommodate changing environments or when population bottlenecks result in fixation of deleterious mutations (Introduction, first paragraph). Yet even under a narrow population bottleneck, fixation of such highly deleterious mutations is unlikely. I think considerable more discussion and references are required to convincingly ground this work in the current understanding of evolution dynamics. Alternatively the authors can put more emphasis on their other, albeit less exciting, conclusions (as the observations that cells are robust to destructive genetic perturbations or new finding in cell biology of polarization).

2) Two fields of research that are highly relevant to this publication receive very little attention in the manuscript and warrant more discussion:

The research of multi-step evolutionary trajectories has been the focus of many previous studies using experimental evolution or directed evolution either in a single protein or in multiple components (e.g., Weinreich et al. Science 2006 and many other papers that followed).

The evolutionary dynamics leading to module reorganization or module take-over. In the Discussion, the authors suggest that module takeover could be a consequence of inactivation of one module (in line with their work) and recruitment of novel proteins that lead to a new module taking over. Are there any references that are compatible with such an adaptation scenario (the existing references only show that is some species, core cellular functions are done by unconventional modules)?

3) The authors state that their study reveals more about the biology of yeast cell polarization (identifying new genes involved). This indeed seems like a valid complementary approach to study this pathway and distill new components. I think this discussion can be further expanded – how do the results compare to previous epistasis maps (SGD website allows to interrogate high throughput E-maps)? What is the potential advantage of identifying relevant genes by monitoring evolutionary change relative to more standard approaches as double deletion screens?

---

## [Author Response]

All referees and the editors agree that the work is exciting. The reproducibility of the evolutionary adaptations, the identity of the mutated genes, and more strikingly, the order of their appearance, are certainly very interesting, especially since a sound epistatic basis for such order is provided.

Yet, both referees questioned the relevance to natural evolution. As clearly put by referee #1: “And it is not even clear whether a Bem1 deletion strain would ever get the chance to evolve in a natural setting as it might be immediately out competed by neighboring wild type cells” and referee #2: “does this starting point, appearance of crippling mutation, reflect the starting point of natural evolutionary scenarios as we understand them?”

To address the issue we added the following text to the Discussion:

One objection to this hypothesis is that it would be very hard to fix a mutation as deleterious as bem1∆. There are ways of countering the objection: severely defective mutations could be fixed in population bottlenecks that accompany speciation; secondly, and maybe more likely, mutations that removed important components could be pleiotropic, offering advantages in novel environments that were similar in magnitude to the costs they impose on strongly conserved core functions; and thirdly, previously selected mutations in the same or related pathways could make the defects associated with the removal of a component less severe.

1) Evolution by gene loss has been written about extensively and the authors should cite some of the previous literature in this area.

Our findings are consistent with previous experimental evolution work which showed that loss of function mutations allow bacteria (26; 65) and yeast (Koschwanez, Foster and Murray, 2013) to adapt to their environments and produce novel phenotypes (62). Adaptation by loss of function mutations has also been observed in natural populations (5; 14; 13; 42; 55).

2) It was not easy to follow exactly how many independent lines were studied. The text says nine mutant lines were evolved, but then ten lines were sequenced. The authors should clarify exactly how many independent lines there were, where they came from, and discuss in more detail which mutations arose independently and which are shared because of common ancestry.

We tried to improve clarity by changing the text at two points:

We sequenced the whole genomes of evolved wild-type lines and bem1∆ lines. From the seven wild-type lines that were left at the end of the evolution experiment, we sequenced five lines. We sequenced a total of ten bem1∆ lines: The eight bem1∆ A-lines that arose from the same starting colony and remained haploid, as described above. In addition, we sequenced two bem1∆ lines that were evolved from two independent different starting colonies in a trial experiment (T2 and T3).

The A-lines shared the same early stop mutation in BEM3, but lines T2 and T3 independently acquired a different early stop mutations in BEM3. The BEM3 mutation in the A-lines (Q61*) was acquired after germination of the spore that acted as their ancestor: cells from the original colony showed the same severe growth defects as freshly germinated bem1∆ spores, whereas engineered bem1∆ bem3∆ cells had a much less severe defect. All five mutations in NRP1 (all in A-lines) were independently acquired early stop mutations, whereas in BEM2 we found a promoter mutation (line T2) and two amino acid substitutions (line A1 and A2), which are radical substitutions at conserved positions.

Reviewer #2:

Two fields of research that are highly relevant to this publication receive very little attention in the manuscript and warrant more discussion: The research of multi-step evolutionary trajectories has been the focus of many previous studies using experimental evolution or directed evolution either in a single protein or in multiple components (e.g., Weinreich et al. Science 2006 and many other papers that followed).

We added the following discussion:

How repeatable is evolution? Experimental evolution has produced a range of answers to this question, from replicate populations that share a small fraction of causal mutations(Koschwanez, Foster and Murray, 2013; [33]; [35]; [11]) to those where mutations occur in the same genes in the same order (6; 23) or repeatedly affect the same amino acids in a protein (43). For multiple substitutions in the same protein, strong epistasis can render most trajectories impossible or unlikely (7; 24; 60). Examining the mutations that gave rise to a particular phenotype in the bacterium, Pseudomonas fluorescens, revealed a hierarchy of pathways: loss of function mutations are most frequent and other pathways, including promoter mutations, gene fusions, and gain of function mutations can only be found when the dominant pathway is blocked (Lind, Farr and Rainey, 2015).

Our work adds to the list of examples of reproducible trajectories and our reconstruction of all possible paths, an exercise that has previously been performed for multiple mutation in a single gene (60), demonstrates why our mutations occurred in a particular order. As far as we know, this is the first example of reproducible trajectories to adaptation to mutations that severely compromise a cellular function, and it will be interesting to see if evolving strains, adapting to severe perturbations in other pathways will also lead to reproducible trajectories.

The evolutionary dynamics leading to module reorganization or module take-over. In the Discussion, the authors suggest that module takeover could be a consequence of inactivation of one module (in line with their work) and recruitment of novel proteins that lead to a new module taking over. Are there any references that are compatible with such an adaptation scenario (the existing references only show that is some species, core cellular functions are done by unconventional modules)?

To address this issue we added the following discussion:

Because of the difficulty of inferring events as ancient as the ones that rearranged the kinetochore of kinetoplastids or made the anaphase promoting complex dispensable in Giardia, it is impossible to say whether it was the loss of genes that inactivated existing pathways, or some other event that triggered the sequence of changes that led particular lineages to use different proteins to perform functions that appeared early in eukaryotic evolution. Despite our inability to reconstruct these processes, there is evidence for individual steps in the process of functional reorganization. These include the loss of widely conserved genes in individual evolutionary lineages (Bergmiller, Ackerman and Silander, 2012; Gourguechon, Holt and Cande, 2013; [12]), the loss of genes present in ancestral species during evolutionary diversification (e.g. the loss of 88 genes in the descent of *S. cerevisiae* from an ancestor that existed 100 Mya ago (Gordon, Byrne and Wolfe, 2009)), and the recruitment of new functions by adaptations that alter the function of existing genes and create genes de novo (44; 45).

3) The authors state that their study reveals more about the biology of yeast cell polarization (identifying new genes involved). This indeed seems like a valid complementary approach to study this pathway and distill new components. I think this discussion can be further expanded – how do the results compare to previous epistasis maps (SGD website allows to interrogate high throughput E-maps)? What is the potential advantage of identifying relevant genes by monitoring evolutionary change relative to more standard approaches as double deletion screens?

We changed to text to discuss those points:

High throughput studies, at least as indicated on the yeast genome website SGD (SGD) have not shown physical or genetic interactions between NRP1 and any of the currently known polarity genes (SGD). BEM1, BEM2 and BEM3 as well as their genetic interactions have been measured and implicated in polarity establishment before (31). However, the lack of information about NRP1 made it impossible to predict the positive effect of the BEM2 deletion on polarity establishment in the absence of BEM1 and BEM3.